# Identifying Latent Concepts and Structures for Generalized Category Discovery

**Boyang Dai** [1]   **Chaoqi Chen** [2]   **Yizhou Yu** [1]

https://github.com/Michael-McQueen/CPF

## Abstract

Generalized Category Discovery (GCD) aims to recognize known classes while autonomously discovering novel ones in open-world settings. However, current approaches primarily focus on designing clustering objectives, often overlooking a critical bottleneck: standard vision backbones yield high-rank, entangled token representations that are ill-suited for unsupervised discovery of latent concepts and structures. In this paper, we propose **C**ompositional **P**rimitive **F**ields (CPF-GCD), a novel representation learning framework that reshapes the feature space to make such latent structure identifiable by enforcing a low-rank compositional organization. Our core hypothesis is that all categories, whether known or novel, can be expressed as compositions and spatial arrangements of a finite set of learnable visual primitives that capture reusable concepts. CPF instantiates this geometric constraint via a spatial field mechanism. Inserted between the backbone and the head, it rewrites noisy patch tokens through low-rank primitive mixtures, effectively decomposing images into reusable atomic parts and their spatial layouts. By explicitly modeling the spatial distribution of primitives, CPF enables novel categories to emerge naturally as new activation patterns over a shared vocabulary. This shifts the focus of representation from merely partitioning global embeddings to constructing a structured and separable primitive field. Extensive experiments demonstrate that CPF serves as a generic, plug-and-play module that consistently boosts performance across diverse GCD baselines, validating that identifying and leveraging low-rank compositional structure is a crucial inductive bias for open-world recognition.

[1]The University of Hong Kong [2]Shenzhen University. Correspondence to: Yizhou Yu <yizhouy@acm.org>.

*Proceedings of the $43^{rd}$ International Conference on Machine Learning*, Seoul, South Korea. PMLR 306, 2026. Copyright 2026 by the author(s).

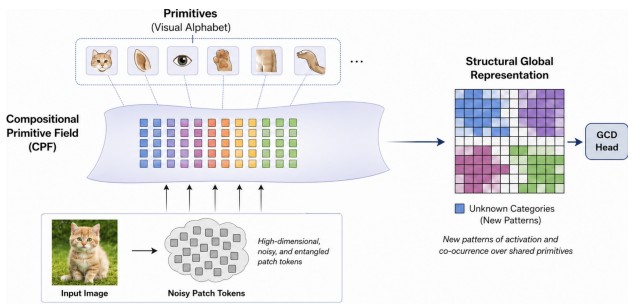

*Figure 1.* **Why structure tokens before discovery?** A compact set of reusable primitives can organize noisy patch tokens before they are consumed by a GCD head, making unknown categories easier to separate as new primitive activation patterns.

## 1. Introduction

Open-world recognition systems are often asked to decide whether an image belongs to a familiar category or to a class that has never been annotated. Generalized category discovery (GCD) (Vaze et al., 2022) formalizes this requirement by training with labeled examples from known classes and unlabeled examples drawn from both known and unknown classes. The desired model must preserve the identity of labeled categories while arranging unlabeled samples from unseen categories into meaningful groups. This setting is deliberately harder than closed-set recognition: the model cannot rely on a fixed label vocabulary, yet it must still produce a representation in which new semantic groups can become visible.

Most GCD methods improve the decision layer attached to a pre-trained visual encoder. Recent systems differ in whether they use clustering objectives (Rastegar et al., 2024b), contrastive criteria (Choi et al., 2024), or prototype-based classifiers (Wen et al., 2023; Cao et al., 2024), but they commonly inherit the same interface: dense visual tokens are collapsed into a global descriptor before a discovery head receives them. When this descriptor is already well organized, such heads can work effectively. When it is not, the head has to separate unknown categories inside a representation that was never built for discovery.

This representation-side bottleneck is particularly visible for modern vision transformers (Park & Kim, 2022). Their global class token is optimized to summarize evidence for

known labels, so it can discard the local evidence that distinguishes fine-grained or previously unseen categories. Patch tokens still contain these local cues, but in the raw backbone space, they appear as a noisy high-dimensional cloud: foreground parts, background textures, and incidental correlations may occupy nearby directions (Izmailov et al., 2022). As a result, discovery heads can suffer from two opposite errors: one category may break into several clusters (Zhang et al., 2021), while different categories may merge through shared context or texture (Krishnakumar et al., 2021; Compton et al., 2023). The issue is therefore not only which loss is used after the backbone, but what geometry the backbone exposes to that loss.

As illustrated in Figure 1, we take a compositional token-organization view of this problem. Instead of asking a global embedding to carry every visual detail, we first express patch tokens through a small set of learnable primitives (Tang et al., 2025; Li et al., 2023). The primitives act as reusable coordinates for local visual evidence, and the patch-to-primitive assignments form a spatial primitive field over the image. Since the primitive set is intentionally compact, the construction introduces a low-rank bottleneck that suppresses redundant variation while retaining recurring semantic structure. Unknown categories do not require entirely new feature axes; they can appear as new compositional activation patterns and spatial arrangements of primitives shared with known categories.

Based on this idea, we introduce **C**ompositional **P**rimitive **F**ields (CPF-GCD), a lightweight module inserted between a vision backbone and a GCD head. CPF learns an image-conditioned primitive basis, refines it through token-primitive interactions, and produces a fused patch-to-primitive assignment used to update the token representation. The module does not replace existing GCD heads or losses. Instead, it changes the representation they receive: before classification or clustering, patch tokens are reorganized through a compact low-rank primitive field and then folded back into the global image representation.

Our contributions are:

- **A representation-side perspective on GCD.** We identify the backbone-to-head interface as a limiting factor in category discovery and argue that token geometry should be structured before applying clustering, contrastive learning, or prototype classification.

- **A primitive-based token organizer.** We propose CPF, a drop-in token organizer that rewrites patch representations through learnable primitives, token-primitive interactions, and adaptive assignment fusion. This gives existing GCD pipelines a compact intermediate space without modifying their heads or objectives.

- **Cross-framework validation and diagnostics.** We

evaluate CPF with representative prototype-based, clustering-based, and contrastive-based GCD methods. The results show consistent gains on known and novel categories, while additional analyses on rank, entropy, attention, and category-number estimation explain how the low-rank primitive field improves discovery.

Overall, this paper shifts part of the GCD design problem from the output head to the representation interface. By reorganizing patch tokens into low-rank primitive fields before discovery, CPF provides a simple way to make pre-trained visual features more suitable for open-world category formation.

## 2. Related Work

**Discovery heads and training signals.** Generalized category discovery (Vaze et al., 2022) studies recognition when annotations cover only a subset of the categories that will appear at training time. A common recipe is to keep a strong visual encoder and improve the module that turns its features into labels or clusters. Some methods use learnable classifiers or prototypes (Wen et al., 2023; Cao et al., 2024; Vaze et al., 2023), while others rely on clustering-style inference (Chiaroni et al., 2023; Zhao et al., 2023; Choi et al., 2024; Rastegar et al., 2024b). Their supervision signals are also diverse, including contrastive objectives (Chen et al., 2020; He et al., 2020), transport-based matching (Fini et al., 2021), consistency regularization (Tarvainen & Valpola, 2017; Sohn et al., 2020), and prompt-based calibration (Zhang et al., 2023; Wang et al., 2024). These designs have substantially advanced the discovery head, but they usually consume the backbone representation as given. CPF is complementary to this line of work: it reorganizes the feature stream before the head sees it, so the same downstream objectives can operate on a cleaner token organization.

**What is passed from the backbone.** Many recent GCD pipelines inherit representations from self-supervised vision transformers, especially DINO-style encoders (Caron et al., 2021; Park & Kim, 2022). Such backbones provide both a global image descriptor and a set of patch tokens. The global descriptor is convenient for classification, yet it can hide the local cues that separate fine-grained or previously unseen classes. Patch tokens expose more evidence, but they also mix foreground parts, textures, and background context in a high-dimensional space (Izmailov et al., 2022). This creates a mismatch: the discovery loss is asked to form semantic groups from features whose local structure has not been explicitly prepared for grouping. Our method targets this interface between encoder and head. Instead of replacing the classifier, clustering rule, or loss, CPF rewrites patch tokens through the compact low-rank primitive fields

and then returns the refined tokens to the ordinary GCD pipeline.

**Reusable units for visual evidence.** Compositional recognition has long suggested that objects and categories can be described through reusable parts, attributes, or local configurations (Biederman, 1987; Lake et al., 2015). Related ideas appear in slot attention and object-centric learning, where visual content is separated into a small set of entities or factors (Locatello et al., 2020; Dai et al., 2026). Generative models also make heavy use of latent factors and token dictionaries to construct visual content (Van Den Oord et al., 2017; Ramesh et al., 2021). CPF adapts this intuition to discriminative discovery rather than generation: its primitives are not output objects, but reusable units that form a low-rank bottleneck for organizing patch evidence. This view is also consistent with the broader observation that compact, well-conditioned representations can improve discrimination (Yerxa et al., 2023; Papyan et al., 2020; Kothapalli et al., 2023). The key difference is operational: CPF implements the compactness as an image-conditioned token rewriting step that can be inserted into existing GCD systems.

## 3. Problem Formulation

**GCD objective.** Let $\mathcal{D}_L = \{(x_i, y_i)\}_{i=1}^{n_L}$ be labeled data from the known label set $\mathcal{Y}_K$, and let $\mathcal{D}_U = \{x_j\}_{j=1}^{n_U}$ be unlabeled data whose samples may come from either known or unknown classes. The unknown label set $\mathcal{Y}_U$ is disjoint from $\mathcal{Y}_K$, and labels in $\mathcal{Y}_U$ are never observed during training. A GCD model must therefore solve two coupled tasks: assign samples from known classes to their correct labels, and partition samples from unknown classes into coherent groups. Existing pipelines usually attach a discovery head $h_\phi$ to a visual backbone $f_\theta$ and optimize supervised and unsupervised objectives jointly. Our method leaves this head and its loss unchanged, and instead modifies the representation delivered to it.

**Backbone-head interface.** For an input image $x$, a transformer backbone produces $N$ patch tokens,

$$\mathbf{X} = [\mathbf{x}_1, \ldots, \mathbf{x}_N]^\top \in \mathbb{R}^{N \times D}, \tag{1}$$

Each $\mathbf{x}_i$ stores the $D$-dimensional feature of one image patch. Standard GCD heads consume a global summary derived from these tokens, such as a class token or pooled feature. This interface is convenient, but it hides how local visual evidence is arranged before discovery. If $\mathbf{X}$ contains many redundant or noisy directions, then clustering and contrastive objectives are applied only after the difficult geometry has already been inherited.

**Primitive reparameterization.** We use a compact token reparameterization before the GCD head. Specifically, CPF

introduces a learnable primitive codebook $\mathbf{P} \in \mathbb{R}^{M \times D}$ and a token-to-primitive assignment matrix $\mathbf{A} \in \mathbb{R}^{N \times M}$, with $M \ll \min\{N, D\}$:

$$\mathbf{X} \approx \mathbf{A}\mathbf{P}. \tag{2}$$

Here, the rows of $\mathbf{P}$ serve as reusable basis elements for local visual evidence, while each row of $\mathbf{A}$ records how a patch token is assigned by that basis. Notably, in CPF, the primitive *field* is instantiated by such *assignments*, which describe how primitives are spatially activated over patch tokens and further refined with each other, as detailed in Section 4. This view does not require assigning a separate representation subspace to every novel class. Instead, known and novel categories may differ by how they compositionally activate and combine the same compact set of primitives.

**Design consequence.** The low-rank bottleneck in Eq. 2 changes where discovery begins. Rather than presenting the GCD head with an unconstrained token cloud, we first rewrite the patch representation through primitive assignment fields and decode it back into a refined token set. The primitive codebook reduces redundant variation, while the assignment matrix preserves patch-level organization, which later instantiates the primitive field used for token rewriting. This gives the downstream GCD objective a more regular representation to cluster or classify, without changing the objective itself.

## 4. Methodology

Following the formulation above, CPF implements the primitive reparameterization in Eq. 2 as a compositional token rewriting block. Given the backbone token matrix $\mathbf{X}$, it constructs an image-conditioned primitive codebook, summarizes token evidence into primitive states, refines these states through lightweight interactions, and decodes the refined primitive information back to the patch-token space. The output has the same dimensionality as the original token matrix, so any GCD head can consume it without architectural changes. As summarized in Figure 2, CPF consists of three stages: (i) image-conditioned primitive codebook construction, which builds the codebook and initial token-to-primitive assignment; (ii) primitive evidence refinement, which accumulates and refines token evidence in the primitive space through token-primitive and primitive-primitive interactions; and (iii) assignment fusion and token rewriting, which fuses assignment cues and decodes the refined primitive mixture back to the patch-token space.

### 4.1. Image-Conditioned Primitive Codebook

Let $\mathbf{X} \in \mathbb{R}^{N \times D}$ denote the patch-token matrix, where $N$ is the number of patches, and $D$ is the feature dimension. The first part of CPF prepares the two ingredients required by

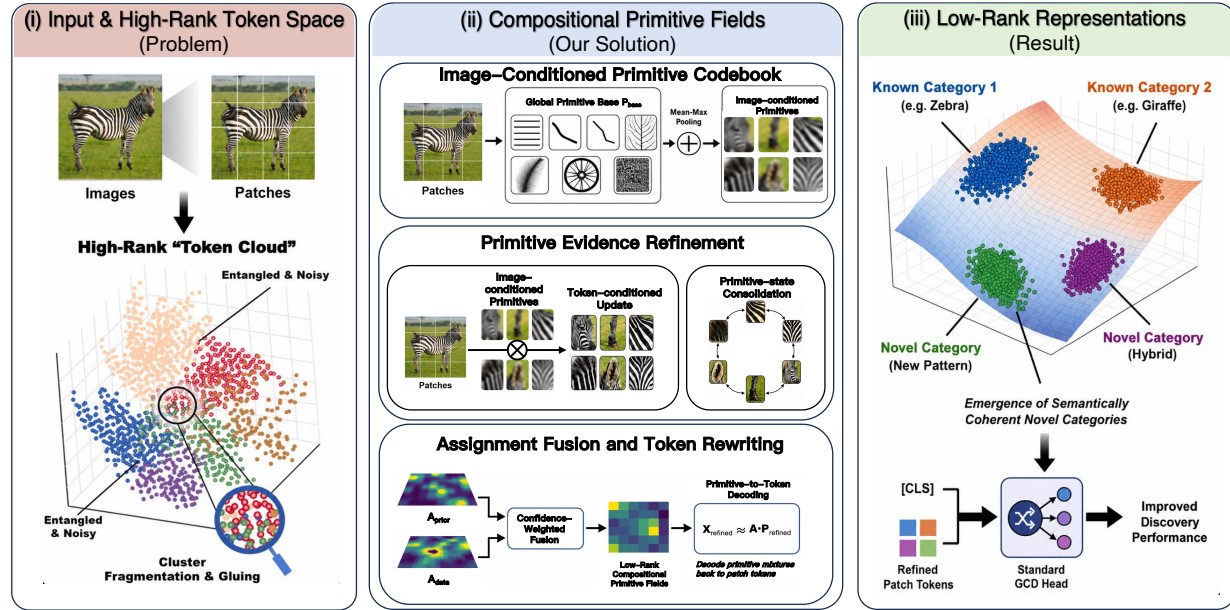

*Figure 2.* Overview of **CPF**. CPF is placed at the backbone-head interface. It rewrites noisy patch tokens through a compact primitive codebook and adaptive assignment fields, then returns a refined token representation to a standard GCD head.

the reparameterization: a small primitive codebook and a provisional token-to-primitive assignment. The codebook supplies the reusable directions, while the assignment specifies which tokens support each direction in the current image.

**Dataset-level codebook.** We maintain a learnable primitive base

$$\mathbf{P}_{\text{base}} \in \mathbb{R}^{M \times D}, \tag{3}$$

where $M$ is the number of primitive slots. The rows of $\mathbf{P}_{\text{base}}$ are shared across the training set and are optimized jointly with the rest of the model. They can capture recurring local evidence such as object parts, textures, or attributes. The value of $M$ controls the capacity of the bottleneck: a larger codebook can represent finer variation, whereas a smaller one forces more aggressive compression. Under the approximation $\mathbf{X} \approx \mathbf{AP}$, $M$ also upper-bounds the effective rank of the rewritten token representation.

**Image-conditioned adaptation.** A fixed codebook is too rigid for images with different poses, scales, and backgrounds. We therefore allow the shared base to receive a small image-conditioned offset. We first summarize $\mathbf{X}$ with a compact context descriptor using mean and max pooled token statistics, and map the descriptor to an additive codebook update:

$$\Delta\mathbf{P} = \Phi\big(\text{MeanMax}(\mathbf{X})\big) \in \mathbb{R}^{M \times D}, \tag{4}$$

where $\Phi$ is a learned linear map followed by reshaping. The codebook used for the current image is

$$\mathbf{P} = \mathbf{P}_{\text{base}} + \Delta\mathbf{P}. \tag{5}$$

This keeps the primitive vocabulary shared, but lets each image slightly adjust the slot descriptors before token assignment.

**Initial token-to-primitive assignment.** Given the adapted codebook, CPF computes a first assignment by comparing tokens with primitives. We project tokens into the primitive space,

$$\hat{\mathbf{X}} = \mathbf{X}W \in \mathbb{R}^{N \times D}, \tag{6}$$

where $W \in \mathbb{R}^{D \times D}$ is trainable, and score token-primitive compatibility by

$$S = \frac{\hat{\mathbf{X}}\mathbf{P}^{\top}}{\sqrt{D}} \in \mathbb{R}^{N \times M}. \tag{7}$$

The initial assignment is

$$\mathbf{A}_{\text{prior}}(i, m) = \frac{\exp(S_{i,m})}{\sum_{j=1}^{N} \exp(S_{j,m})}, \quad \mathbf{A}_{\text{prior}} \in \mathbb{R}^{N \times M}. \tag{8}$$

We normalize over tokens for each primitive, so every primitive slot distributes a unit amount of support across the image. This column-wise normalization makes tokens compete for each primitive slot. As a consequence, primitives tend to concentrate on informative regions instead of spreading uniformly over all patches. We use $\mathbf{A}_{\text{prior}}$ as the prior assignment for the next step.

### 4.2. Primitive Evidence Refinement

The codebook stage provides $\mathbf{P}$ and $\mathbf{A}_{\text{prior}}$. Together they give a first approximation $\mathbf{X} \approx \mathbf{A}_{\text{prior}}\mathbf{P}$. This approximation is useful but still shallow: it is obtained from pairwise

affinity and does not yet let primitives collect broader token evidence or coordinate with other primitives. CPF therefore performs relational refinement in the primitive space before decoding back to tokens.

**Token evidence pooling.** We first gather token evidence into $M$ primitive states using the prior assignment:

$$\mathbf{P}^0 = \mathbf{A}_{\text{prior}}^\top \mathbf{X} \in \mathbb{R}^{M \times D}. \qquad (9)$$

The $m$-th row of $\mathbf{P}^0$ is the token summary assigned to primitive $m$. This converts the potentially large token set into a compact collection of primitive states, which is cheaper and more stable to reason over.

**Token-conditioned update.** The pooled states are then allowed to query the original tokens once more. We project primitive states and tokens into attention subspaces:

$$Q_{\text{TP}} = \mathbf{P}^0 W_q, \quad K_{\text{TP}} = \mathbf{X} W_k, \quad V_{\text{TP}} = \mathbf{X} W_v. \qquad (10)$$

The primitive-to-token interaction is

$$\mathbf{W}_{\text{TP}} = \text{softmax}\left(\frac{Q_{\text{TP}} K_{\text{TP}}^\top}{\sqrt{d_h}}\right), \qquad (11)$$

and the primitive states are updated as

$$\mathbf{P}^1 = \mathbf{P}^0 + \gamma\, \mathbf{W}_{\text{TP}} V_{\text{TP}}. \qquad (12)$$

This second pass lets each primitive recover useful evidence that may have been under-weighted by the initial assignment. The residual scale $\gamma$ controls the strength of this token-conditioned correction.

**Primitive-state consolidation.** Finally, primitives exchange information with each other to remove redundancy and make the codebook states more coherent. Starting from $\mathbf{P}^1$, we compute

$$Q_{\text{PP}} = \mathbf{P}^1 W_q', \quad K_{\text{PP}} = \mathbf{P}^1 W_k', \quad V_{\text{PP}} = \mathbf{P}^1 W_v'. \qquad (13)$$

The primitive-to-primitive kernel is

$$\mathbf{W}_{\text{PP}} = \text{softmax}\left(\frac{Q_{\text{PP}} K_{\text{PP}}^\top}{\sqrt{d_h}}\right), \qquad (14)$$

which gives the final refined primitive states

$$\mathbf{P}^2 = \mathbf{W}_{\text{PP}} V_{\text{PP}}. \qquad (15)$$

In practice, both primitive-to-token and primitive-to-primitive interactions use $H$ interaction heads. For compact notation, Eqs. 11–15 omit the head index and are written as the concatenated and linearly projected multi-head messages in $\mathbb{R}^{M \times D}$.

Therefore, the full refinement path is

$$\mathbf{X} \xrightarrow{\text{aggregation}} \mathbf{P}^0 \xrightarrow{\text{refinement}} \mathbf{P}^1 \xrightarrow{\text{contraction}} \mathbf{P}^2 \qquad (16)$$

which turns raw token evidence into a compact set of refined primitive states.

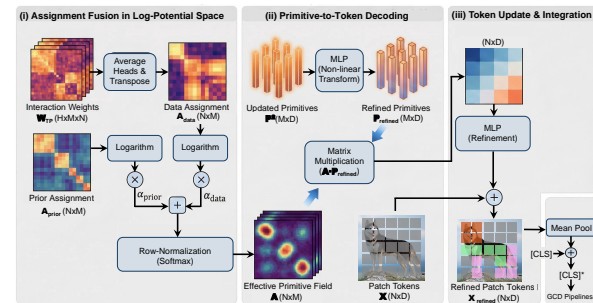

*Figure 3.* Assignment fusion and token rewriting in Section 4.3.

### 4.3. Assignment Fusion and Token Rewriting

The refinement stage produces $\mathbf{P}^2 \in \mathbb{R}^{M \times D}$. To update the original patch tokens, CPF also needs a final token-to-primitive assignment. As illustrated in Figure 3, we derive one assignment from the initial compatibility scores and another from the refinement dynamics, fuse them, and decode primitive messages back to tokens.

**Readout from refinement dynamics.** The interaction in Eq. 11 produces multi-head weights $W_{\text{TP}}^{(h)} \in \mathbb{R}^{M \times N}$. Although these weights are used to update primitives from tokens, they also indicate which tokens actually contributed to each primitive during refinement. We convert this evidence into a token-to-primitive assignment by transposing the interaction map, averaging over heads, and normalizing over primitives:

$$\mathbf{A}_{\text{data}}^{(i,m)} = \text{Softmax}_m\left(\frac{1}{H} \sum_{h=1}^{H} W_{\text{TP}}^{(h,m,i)}\right), \qquad (17)$$

Here, data assignment $\mathbf{A}_{\text{data}} \in \mathbb{R}^{N \times M}$ records how strongly token $i$ is linked to primitive $m$ after the refinement step. Compared with $\mathbf{A}_{\text{prior}}$, it is less tied to the initial score and more tied to the measured token-primitive exchange.

**Confidence-weighted fusion.** The two assignments play different roles. $\mathbf{A}_{\text{prior}}$ is stable because it comes directly from the adapted codebook and local token scores; $\mathbf{A}_{\text{data}}$ is adaptive because it comes from the refinement trajectory. CPF combines them in log-potential space using two learnable scalar weights:

$$\mathbf{Z} = \alpha_{\text{prior}} \log \mathbf{A}_{\text{prior}} + \alpha_{\text{data}} \log \mathbf{A}_{\text{data}}. \qquad (18)$$

After exponentiation and row normalization, the final assignment is

$$\mathbf{A}(i,m) = \frac{\exp\left(\mathbf{Z}(i,m)\right)}{\sum_{m'=1}^{M} \exp\left(\mathbf{Z}(i,m')\right)}. \qquad (19)$$

**Primitive-to-token decoding.** We transform the refined primitives with a lightweight MLP

$$\mathbf{P}_{\text{refined}} = \text{MLP}(\mathbf{P}^2) \in \mathbb{R}^{M \times D}, \qquad (20)$$

and use $\mathbf{A}$ to decode primitive messages to the patch-token space:

$$\mathbf{X}_{\text{refined}} = \mathbf{X} + \text{MLP}(\mathbf{A} \cdot \mathbf{P}_{\text{refined}}). \qquad (21)$$

The residual connection keeps the original backbone signal, while the decoded primitive mixture injects the compact, denoised structure learned by CPF. The output $\mathbf{X}_{\text{refined}}$ remains an $N \times D$ token matrix, so it can replace $\mathbf{X}$ wherever a standard GCD pipeline expects patch tokens.

**Integration into GCD pipelines.** For image-level discovery heads, we update the class token by pooling the refined patch tokens:

$$[\textbf{CLS}]^* = [\textbf{CLS}] + \text{Mean}(\mathbf{X}_{\text{refined}}). \qquad (22)$$

The enriched representation is then passed to the original GCD head, whether it is prototype-based, clustering-based, or contrastive. Thus CPF changes only the representation interface and does not require modifying the downstream loss or head design.

## 5. Experiments

Through comprehensive experiments, we seek to answer three core questions: (1) **Effectiveness.** Does CPF-GCD consistently improve discovery performance across diverse benchmarks? (2) **Generality.** Is our framework a truly generic, plug-and-play module that boosts various existing GCD baselines? (3) **Mechanism.** What drives the performance gains? Is it the assignments, or their dynamic fusion?

### 5.1. Experimental Setup

**Datasets and Evaluation Protocols.** We evaluate CPF-GCD on both coarse- and fine-grained benchmarks, including fine-grained domains (CUB-200 (Wah et al., 2011), Stanford Cars (Krause et al., 2013), FGVC Aircraft (Maji et al., 2013)) and coarse-grained generic classification (CIFAR-10 (Krizhevsky et al., 2009), CIFAR-100, ImageNet-100 (Deng et al., 2009)). In experiments, we strictly follow the splits and protocols established in (Vaze et al., 2021; 2022). We report clustering accuracy (ACC) on *known/old*, *novel/new*, and *all* classes, aligning predictions via the Hungarian algorithm (Kuhn, 1955).

**Implementation Details.** We implement the proposed CPF-GCD as a lightweight, plug-and-play module designed to seamlessly integrate with diverse GCD architectures. Consistent with previous works (Vaze et al., 2022; Zhang et al., 2023; Pu et al., 2023), we employ a frozen DINO ViT-B/16 (Caron et al., 2021) pre-trained on ImageNet-1K (Deng et al., 2009) as the feature extractor, utilizing the sequence of patch tokens as input to our model. For

CPF, we set the number of learnable primitives $M$ and interaction heads $H$ to 12 on fine-grained datasets, and to 16 on coarse-grained datasets. To rigorously evaluate the generality of our approach, we strictly adhere to the original training hyperparameters and optimization schedules of each host baseline, introducing no method-specific tuning.

**Baselines.** We select mainstream GCD paradigms as the plug-and-play target schemes for CPF, including contrastive learning-based methods (CMS (Choi et al., 2024), SelEx (Rastegar et al., 2024b)) and prototype learning-based methods (SimGCD (Wen et al., 2023), LegoGCD (Cao et al., 2024)). For comprehensive comparison, we further include GCD (Vaze et al., 2022), $\mu$GCD (Vaze et al., 2024), PromptCAL (Zhang et al., 2023), DCCL (Pu et al., 2023), InfoSieve (Rastegar et al., 2024a), AMEND (Banerjee et al., 2024), PIM (Chiaroni et al., 2023), ProtoGCD (Ma et al., 2025), APL (Dai et al., 2025) with SimGCD and ConGCD (Tang et al., 2025) with SPTNet (Wang et al., 2024) in our evaluation.

### 5.2. Results

**Results on fine-grained datasets.** Table 1 validates the efficacy of CPF-GCD across three fine-grained benchmarks. As a plug-and-play module, CPF-GCD consistently outperforms four baselines. Notably, it boosts average accuracy by 4.75% on Stanford-Cars, with a 6.00% gain on novel classes across all baselines. Significant gains in novel class discovery reach **8.1%** with LegoGCD, and even with the strong SelEx baseline, CPF-GCD achieves a peak accuracy of 79.8% on CUB-200. These results confirm that modeling the low-rank compositional primitives effectively resolves fine-grained differences that global features struggle with.

**Results on coarse-grained datasets.** To ensure CPF-GCD's effectiveness on global tasks, we evaluate it on CIFAR-10, CIFAR-100, and ImageNet-100 (Table 2). Unlike specialized fine-grained models, CPF-GCD remains strong across coarse-grained benchmarks. On ImageNet-100, it boosts novel class discovery by 2.03% on average, with a peak of **3.1%** with SelEx. While slight fluctuations in known class accuracy (*e.g.*, with LegoGCD) occur, this suggests that CPF reallocates part of the representation capacity from known-class specialization to novel-cluster formation. These results show that CPF-GCD enriches the feature space with structural details, complementing global semantics for generic object classification.

### 5.3. Ablation Study

**Ablation on the number of primitives.** As shown in Figure 4, we observe a consistent performance plateau across fine-grained benchmarks. Our method achieves optimal and stable results within the range of $M \in [12, 16]$, revealing

*Table 1.* Performance comparison on the fine-grained semantic shift benchmark. The best and runner-up results are marked in bold black text and underlined, respectively. For △, positive growth is indicated by bold green text, while negative growth is shown in standard red text, explicitly marked with a minus sign.

| Method | CUB-200 | | | FGVC-Aircraft | | | Stanford-Cars | | | Average | | |
|---|---|---|---|---|---|---|---|---|---|---|---|---|
| | All | Old | New | All | Old | New | All | Old | New | All | Old | New |
| GCD | 51.3 | 56.6 | 48.7 | 45.0 | 41.1 | 46.9 | 39.0 | 57.6 | 29.9 | 45.1 | 51.8 | 41.8 |
| PromptCAL | 62.9 | 64.4 | 62.1 | 52.2 | 52.2 | 52.3 | 50.2 | 70.1 | 40.6 | 55.1 | 62.2 | 51.7 |
| AMEND | 64.9 | 75.6 | 59.6 | 52.8 | 61.8 | 48.3 | 56.4 | 73.3 | 48.2 | 58.0 | 70.2 | 52.0 |
| $\mu$GCD | 65.7 | 68.0 | 64.6 | 53.8 | 55.4 | 53.0 | 56.5 | 68.1 | 50.9 | 58.7 | 63.8 | 56.2 |
| ProtoGCD | 63.2 | 68.5 | 60.5 | 56.8 | 62.5 | 53.9 | 53.8 | 73.7 | 44.2 | 57.9 | 68.2 | 52.9 |
| InfoSieve | 69.4 | 77.9 | 65.2 | 56.3 | 63.7 | 52.5 | 55.7 | 74.8 | 46.4 | 60.5 | 72.1 | 54.7 |
| APL | 64.5 | 68.1 | 62.1 | 56.6 | 60.2 | 54.8 | **60.1** | 77.6 | 51.2 | 60.4 | 68.6 | 56.0 |
| ConGCD | 68.1 | 68.5 | 67.8 | 59.7 | 61.3 | **59.2** | 59.1 | **79.0** | 49.8 | 62.3 | 69.6 | 58.9 |
| SimGCD | 61.2 | 65.8 | 58.9 | 54.5 | 59.3 | 52.1 | 54.6 | 72.8 | 45.7 | 56.8 | 66.0 | 52.2 |
| + CPF-GCD | 63.9 | 65.5 | 63.1 | 55.8 | 59.1 | 54.2 | 57.6 | 73.7 | 49.8 | 59.1 | 66.1 | 55.7 |
| | **2.7** | -0.3 | **4.2** | **1.3** | -0.2 | **2.1** | **3.0** | **0.9** | **4.1** | **2.3** | **0.1** | **3.5** |
| LegoGCD | 61.9 | 71.9 | 56.9 | 54.6 | 62.7 | 50.6 | 53.7 | 72.2 | 44.9 | 56.7 | 68.9 | 50.8 |
| + CPF-GCD | 66.0 | 73.5 | 62.0 | 56.3 | 62.3 | 53.2 | 59.7 | 73.6 | **53.0** | 60.7 | 69.8 | 56.1 |
| | **4.1** | **1.6** | **5.1** | **1.7** | -0.4 | **2.6** | **6.0** | **1.4** | **8.1** | **4.0** | **0.9** | **5.3** |
| CMS | 65.7 | 75.8 | 60.7 | 50.8 | 62.1 | 45.1 | 51.3 | 73.5 | 40.6 | 55.9 | 70.5 | 48.8 |
| + CPF-GCD | 66.8 | 74.8 | 62.9 | 51.5 | 60.4 | 47.0 | 57.9 | 78.2 | 48.0 | 58.7 | 71.1 | 52.6 |
| | **1.1** | -1.0 | **2.2** | **0.7** | -1.7 | **1.9** | **6.6** | **4.7** | **7.4** | **2.8** | **0.6** | **3.8** |
| SelEx | 75.6 | 77.3 | 74.7 | 61.1 | 68.7 | 57.3 | 55.5 | 77.6 | 44.8 | 64.1 | 74.5 | 58.9 |
| + CPF-GCD | **79.8** | **80.5** | **79.4** | **61.8** | **69.0** | 58.1 | 58.9 | **79.0** | 49.2 | **66.8** | **76.2** | **62.2** |
| | **4.2** | **3.2** | **4.7** | **0.7** | **0.3** | **0.8** | **3.4** | **1.4** | **4.4** | **2.7** | **1.7** | **3.3** |
| Avg. △ | **3.03** | **0.88** | **4.05** | **1.10** | -0.50 | **1.85** | **4.75** | **2.10** | **6.00** | **2.95** | **0.83** | **3.98** |

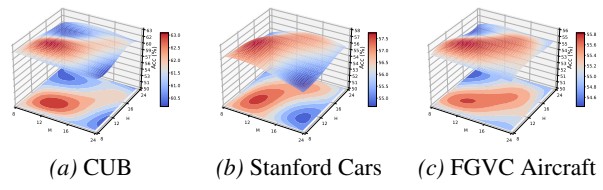

*(a)* CUB     *(b)* Stanford Cars     *(c)* FGVC Aircraft

*Figure 4.* Hyperparameter sensitivity of the number of primitives ($M$) and interaction heads ($H$).

that CPF-GCD is robust to changes in primitive capacity and does not require dataset-specific fine-tuning, validating its generalizability for diverse open-world scenarios. A core design philosophy of CPF-GCD is to *minimize the dependency on sensitive hyperparameters*, ensuring its utility as a practical, plug-and-play solution. To this end, we couple the primitive capacity $M$ with the number of attention heads $H$ (setting $M = H$) in practice.

**Ablation on components.** We conduct an ablation study (Table 3) to evaluate our method's key components. First, we replace the refined aggregation Mean($\mathbf{X}_{\text{refined}}$) with a global average Mean($\mathbf{X}$) (denoted as *w/ patches*), which shows marginal improvement over SimGCD but underperforms CPF-GCD, emphasizing the importance of primitive field token rewriting. Removing the data-assignment ($\mathbf{A}_{\text{data}}$) causes the sharpest drop on CUB-200 (from 63.1% to 60.2%), while excluding the prior-assignment ($\mathbf{A}_{\text{prior}}$) degrades novel class accuracy on Stanford Cars to 48.9%.

Finally, omitting the learnable fusion gates $\alpha$ results in suboptimal performance, confirming the need for dynamic arbitration between prior stability and data-driven specificity.

## 6. Further Empirical Analysis

We analyze CPF from three perspectives: geometric properties, semantic interpretability, and computational efficiency.

**CPF Decreases Von Neumann Entropy.** We use von Neumann Entropy (VNE) (Boes et al., 2019) to examine the geometric transformation induced by CPF-GCD. VNE is calculated from the autocorrelation matrix $\mathcal{R}$ of the feature space, and it measures the uniformity of spectral energy distribution. High entropy typically indicates isotropic distribution or unstructured redundancy, which can hinder unsupervised learning in GCD by entangling semantic cues with high-frequency noise. As shown in Figure 5, CPF reduces both VNE and effective rank compared to SelEx, leading to better discovery performance. This suggests that CPF filters out redundant dimensions and reorganizes high-rank token features into a compact, structured primitive field. We interpret this entropy reduction as a *spectral purification* process that refines the representation into the semantic evidence essential for distinguishing categories.

**CPF Offers Precise Representation Distribution Estimation.** Table 4 shows that CPF-GCD provides much more accurate estimates of unseen category cardinality compared

*Table 2.* Performance comparison on the coarse-grained classification benchmark.

| Method | CIFAR-10 | | | CIFAR-100 | | | ImageNet-100 | | | Average | | |
|---|---|---|---|---|---|---|---|---|---|---|---|---|
| | All | Old | New | All | Old | New | All | Old | New | All | Old | New |
| GCD | 91.5 | **97.9** | 88.2 | 73.0 | 76.2 | 66.5 | 74.1 | 89.8 | 66.3 | 79.5 | 88.0 | 73.7 |
| PIM | 94.7 | 97.4 | 93.3 | 78.3 | 84.2 | 66.5 | 83.1 | 95.3 | 77.0 | 85.4 | 92.3 | 78.9 |
| PromptCAL | **97.9** | 96.6 | **98.5** | 81.2 | 84.2 | 75.3 | 83.1 | 92.7 | 78.3 | 87.4 | 91.2 | 84.0 |
| DCCL | 96.3 | 96.5 | 96.9 | 75.3 | 76.8 | 70.2 | 80.5 | 90.5 | 76.2 | 84.0 | 87.9 | 81.1 |
| ProtoGCD | 97.3 | 95.3 | 98.2 | 81.9 | 82.9 | 80.0 | 84.0 | 92.2 | 79.9 | 87.7 | 90.1 | 86.0 |
| InfoSieve | 94.8 | 97.7 | 93.4 | 78.3 | 82.2 | 70.5 | 80.5 | 93.8 | 73.8 | 84.5 | 91.2 | 79.2 |
| APL | 97.1 | 94.9 | 98.2 | 80.9 | 81.6 | 79.5 | 83.2 | 92.6 | 78.5 | 87.1 | 89.7 | 85.4 |
| ConGCD | 97.4 | 95.2 | **98.5** | 82.5 | 85.9 | 77.3 | 85.9 | 93.4 | 82.5 | 88.6 | 91.5 | 86.1 |
| SimGCD | 96.5 | 96.1 | 96.7 | 80.1 | 82.1 | 76.0 | 83.2 | 94.0 | 77.8 | 86.6 | 90.7 | 83.5 |
| + CPF-GCD | 97.4 | 95.5 | 98.3 | 80.9 | 82.0 | 78.8 | 84.2 | 92.7 | 80.0 | 87.5 | 90.1 | 85.7 |
| | **0.9** | -0.6 | **1.6** | **0.8** | -0.1 | **2.8** | **1.0** | -1.3 | **2.2** | **0.9** | -0.6 | **2.2** |
| LegoGCD | 96.8 | 96.1 | 97.2 | 81.9 | 83.5 | 78.7 | **86.3** | 94.5 | 82.1 | 88.3 | 91.4 | 86.0 |
| + CPF-GCD | 97.5 | 95.8 | 98.3 | 82.8 | 83.3 | **81.9** | 86.1 | 93.1 | **82.6** | **88.8** | 90.7 | **87.6** |
| | **0.7** | -0.3 | **1.1** | **0.9** | -0.2 | **3.2** | -0.2 | -1.4 | **0.5** | **0.5** | -0.7 | **1.6** |
| CMS | 95.2 | 96.9 | 94.4 | 82.4 | **86.0** | 75.3 | 83.1 | 94.1 | 77.6 | 86.9 | 92.3 | 82.4 |
| + CPF-GCD | 96.2 | 97.0 | 95.7 | **83.2** | 85.8 | 78.1 | 85.1 | 95.4 | 79.9 | 88.2 | 92.7 | 84.6 |
| | **1.0** | **0.1** | **1.3** | **0.8** | -0.2 | **2.8** | **2.0** | **1.3** | **2.3** | **1.3** | **0.4** | **2.2** |
| SelEx | 95.5 | 97.1 | 94.6 | 82.1 | 85.1 | 76.2 | 83.3 | 94.4 | 77.7 | 87.0 | 92.2 | 82.8 |
| + CPF-GCD | 96.8 | 97.7 | 96.3 | 82.0 | 84.4 | 77.3 | 85.5 | 94.7 | 80.8 | 88.1 | 92.3 | 84.8 |
| | **1.3** | **0.6** | **1.7** | -0.1 | -0.7 | **1.1** | **2.2** | **0.3** | **3.1** | **1.1** | **0.1** | **2.0** |
| Avg. △ | **0.98** | -0.05 | **1.43** | **0.60** | -0.30 | **2.48** | **1.25** | -0.28 | **2.03** | **0.95** | -0.20 | **2.00** |

*Table 3.* Ablations on components.

| Components | CUB-200 | | | Aircraft | | | S-Cars | | |
|---|---|---|---|---|---|---|---|---|---|
| | All | Old | New | All | Old | New | All | Old | New |
| SimGCD | 61.2 | 65.8 | 58.9 | 54.5 | 59.3 | 52.1 | 54.6 | 72.8 | 45.7 |
| w/ patches | 62.1 | 65.8 | 60.3 | 54.8 | 59.3 | 52.6 | 55.4 | 70.9 | 49.1 |
| + CPF-GCD | 63.9 | 65.5 | 63.1 | 55.8 | 59.1 | 54.2 | 57.6 | 73.7 | 49.8 |
| w/o $\alpha$ | 63.2 | 64.3 | 62.7 | 55.0 | 58.2 | 53.5 | 56.6 | 74.1 | 48.1 |
| w/o $A_{prior}$ | 62.6 | 62.3 | 62.7 | 54.9 | 57.9 | 53.4 | 56.7 | 72.6 | 48.9 |
| w/o $A_{data}$ | 62.2 | 66.2 | 60.2 | 55.1 | 58.0 | 53.7 | 57.1 | 73.2 | 49.3 |

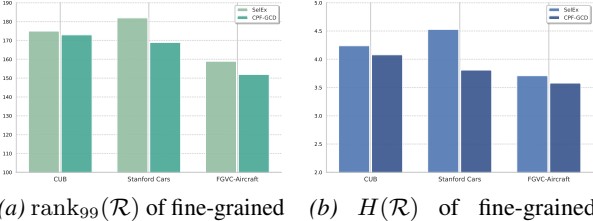

*(a)* $\mathrm{rank}_{99}(\mathcal{R})$ of fine-grained datasets

*(b)* $H(\mathcal{R})$ of fine-grained datasets

*Figure 5.* Comparison between $\mathrm{rank}_{99}(\mathcal{R})$ and $H(\mathcal{R})$. Here, $\mathrm{rank}_{99}(\mathcal{R})$ is the count of the largest eigenvalues needed to account for 99% of the total eigenvalue energy.

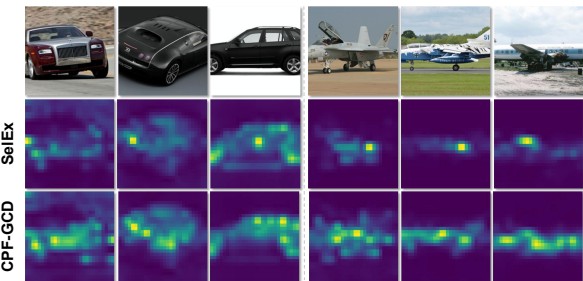

*Figure 6.* Visualization of attention maps.

*Table 4.* Estimated number and error rate.

| Method | CIFAR-100 | | IN-100 | | CUB-200 | | S-Cars | |
|---|---|---|---|---|---|---|---|---|
| | $|\mathcal{Y}_u|$ | Er(%) | $|\mathcal{Y}_u|$ | Er(%) | $|\mathcal{Y}_u|$ | Er(%) | $|\mathcal{Y}_u|$ | Er(%) |
| Ground Truth | 100 | – | 100 | – | 200 | – | 196 | – |
| GCD | 100 | 0 | 109 | 9 | 231 | 15.5 | 230 | 17.3 |
| DCCL | 146 | 46 | 129 | 29 | 172 | 14 | 192 | 2.04 |
| CMS | 92 | 8 | 105 | 5 | 164 | 18 | 153 | 21.9 |
| + CPF-GCD | 96 | 4 | 97 | 3 | 177 | 11.5 | 160 | 18.4 |

to CMS. CPF-GCD reduces estimation errors by half on CIFAR-100 and cuts the error margin from 18% to 11.5% on CUB-200. This improvement highlights how grounding the representation space in a low-rank compositional field mitigates geometric confusion and prevents semantic clusters from merging incorrectly.

**CPF Reshapes the Model's Attention.** By constraining patch tokens to a compact set of primitives, CPF-GCD transforms the attention mechanism from raw pixel correlations

to interactions between primitive tokens. As shown in Figure 6, the baseline exhibits diffuse attention patterns, failing to separate the object of interest from irrelevant context. In contrast, CPF-GCD yields attention maps that better cover the main foreground objects, such as car bodies and aircraft fuselages or wings, while suppressing irrelevant background regions. This suggests that the learned primitive field encourages patch tokens to align with reusable object-level structures, thereby improving the signal-to-noise ratio for more robust category discovery.

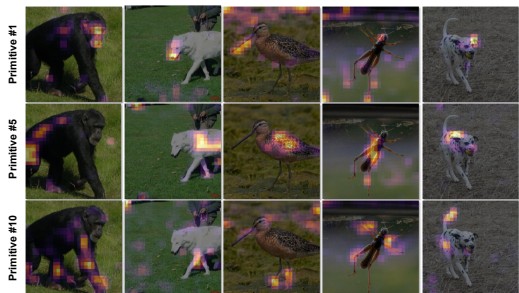

*Figure 7.* Visualization of the primitives.

*Table 5.* Computational overhead.

| Models | Params (M) | Training Time (s) | Inference Time (s) |
|---|---|---|---|
| SimGCD | 92.10 | 27.87 | 10.61 |
| + CPF-GCD | 99.62 | 30.10 | 11.25 |
| △ | + 7.5 | + 8% | + 6% |

**Primitives are Consistent Across Instances.** To validate the semantic consistency of the learned primitive field, we visualize the spatial activation of primitives on ImageNet-100. As shown in Figure 7, the primitives consistently capture cross-category semantics. For example, Primitive #1 focuses on head regions, Primitive #5 attends to the main torso, and Primitive #10 associates with limbs and tails. These patterns indicate that primitives act as a visual alphabet of reusable parts, enabling the model to recognize novel categories as combinations of familiar structural elements, enhancing generalization in the open world.

**Minimal Computational Overhead.** As shown in Table 5, CPF adds approximately 7.5M of parameters to the overall model, with minimal impact on performance. Training time increases by only 8%, and inference time remains competitive. This efficiency comes from performing token organization through a compact low-rank primitive field, where interactions are mediated by the primitives rather than relying only on the original high-rank patch-token space. CPF-GCD offers consistent accuracy gains while being a lightweight and practical module.

## 7. Conclusion

In this work, we identified the unstructured, high-rank geometry of standard backbone representations as a fundamental bottleneck hindering Generalized Category Discovery and proposed **C**ompositional **P**rimitive **F**ields (CPF-GCD) to fundamentally reshape this latent space. By explicitly modeling visual data with a compact primitive codebook and spatial token-to-primitive assignment fields, CPF-GCD effectively filters out spurious noise, enabling novel categories to emerge naturally as distinct activation patterns and spatial

configurations over a shared primitive vocabulary. Extensive experiments across diverse benchmarks confirm that CPF serves as a potent, plug-and-play inductive bias, delivering consistent performance gains, accurate cardinality estimation, and minimal computational overhead. Our findings demonstrate that enforcing structural compositionality is a critical missing link in open-world recognition, providing a new direction for future research on dynamic primitive field construction for ever-expanding category spaces.

## Acknowledgments

Chaoqi Chen is supported by the National Natural Science Foundation of China Excellent Young Scientists Fund Program (Overseas) and the Guangdong Provincial Young Topnotch Talent Program.

## Impact Statement

This paper presents work whose goal is to advance the field of Machine Learning, particularly generalized category discovery and open-world visual recognition. While our method may help reduce annotation costs and improve the adaptability of visual recognition systems to emerging categories, we do not identify any specific societal consequences that require further discussion here.

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
