# OpenReview forum: "Identifying Latent Concepts and Structures for Generalized Category Discovery"
_ICML.cc/2026/Conference — ICML 2026 regular_

### Official Review · Reviewer_3VZj · 2026-03-01

**Soundness:** 3
**Presentation:** 4
**Significance:** 3
**Originality:** 3
**Overall Recommendation:** 6
**Confidence:** 5

**Summary:**

This paper proposes Compositional Primitive Fields for Generalized Category Discovery (CPF-GCD), motivated by the observation that standard backbone representations used in GCD tend to be geometrically high-rank and weakly structured, which can hinder reliable separation of known vs. novel categories. The method introduces a set of data-driven primitives and performs message passing and token aggregation in a primitive field space, effectively shifting interactions from dense patch-to-patch attention to more structured primitive-to-token reasoning. The proposed module is designed to be plug-and-play on top of existing GCD pipelines. Experiments on both coarse-grained and fine-grained benchmarks show consistent improvements, with particularly strong gains on novel-class discovery performance.

**Compliance With Llm Reviewing Policy:**

Affirmed.

**Final Justification:**

Thanks for the reply. I think my concerns have been addressed, and I will raise the score accordingly.

**Key Questions For Authors:**

1.Why does CPF-GCD tend to cause a decline in recognition accuracy for known categories, a trend that appears even more pronounced on coarse-grained datasets?\
2.Across both coarse-grained and fine-grained datasets, the performance gains of CPF-GCD when applied to SelEx appear generally more stable compared to other methods. Does this suggest that although CPF-GCD is designed as a plug-and-play module, its effectiveness is heavily influenced by the preceding modules or the specific host framework?

**Limitations:**

yes

**Strengths And Weaknesses:**

Strengths\
1.Strong Motivation and Insight: The paper provides a very insightful diagnosis of why current GCD methods struggle. Pointing out the discrepancy between the high-rank, entangled nature of standard vision transformer representations and the low-rank requirements for unsupervised concept discovery is a significant contribution to the community.\
2. Methodological Innovation: The proposed Compositional Primitive Field (CPF) is highly innovative and elegant. The design choices—specifically the adaptive primitive generator and the interaction/consolidation mechanisms—are theoretically sound and well-motivated for forcing the network to learn a low-rank compositional manifold.\
3. Convincing Empirical Results: The experimental validation is thorough. The CPF framework demonstrates consistent and substantial performance improvements across multiple standard GCD benchmarks. The ablation studies effectively validate the contribution of each proposed component.\
4. Clarity and Presentation: The paper is exceptionally well-written and easy to follow. The mathematical formulations are clear, and the visual illustrations of the high-rank vs. low-rank spaces greatly aid in understanding the core intuition behind the method.\
Weaknesses\
1.Sensitivity to Hyperparameters (Number of Primitives): While the framework is effective, the performance likely depends on the predefined number of latent primitives (e.g., K). The paper could benefit from a more exhaustive sensitivity analysis on how the choice of K impacts performance across datasets with vastly different scales and granularities. A discussion on how to dynamically determine or heuristically initialize K would strengthen the paper.\
2. Computational Overhead Analysis: The introduction of the primitive generator and the token-primitive cross-attention mechanisms inevitably introduces additional computational complexity. A more detailed profiling of the training and inference overhead (e.g., FLOPs, memory footprint, wall-clock time) compared to the standard GCD baselines is needed to fully contextualize the trade-off between performance and efficiency.\
3. Generalization in Extreme Domain Shifts: The paper demonstrates good performance on standard benchmarks, but it would be interesting to see or discuss the limitations of the compositional primitives when the novel classes exhibit extreme semantic or domain shifts from the labeled known classes. Are the learned primitives truly universal enough to compose completely out-of-distribution concepts? Expanding on the failure cases would provide valuable insights.

---

> ### Author Rebuttal · Authors · 2026-03-30
>
> We sincerely thank the reviewer for the constructive feedback. We appreciate the opportunity to address the concerns regarding hyperparameter sensitivity, computational overhead, generalization to extreme domain shifts, and the trade-offs in known-category accuracy.
>
> > **Q1: Sensitivity to number of primitives (K)?**
> - **Cause:** K controls the expressiveness of the primitive basis. **Too small** causes under-composition (distinct parts merged). **Too large** yields an over-complete basis that can reconstruct high-rank noise and weaken the denoising effect.
> - **Empirical guidance:** A **moderate-range K** is stable across datasets in our experiments. We recommend a short grid search across a few K values spanning an order of magnitude; a brief validation sweep is typically sufficient.
> - **Heuristic initialization:** Set K proportional to the expected number of recurring mid-level parts in the domain (smaller for generic natural images, larger for very fine-grained tasks).
> - **Future work:** We plan to add automatic selection (e.g., sparsity-regularized pruning or lightweight relevance scoring) to reduce manual tuning.
>
> > **Q2: Computational overhead?**
> - **Parameter cost:** **Negligible**—CPF adds only a tiny number of parameters relative to standard backbones.
> - **Runtime & memory:** Training overhead is **modest** (mainly due to OT-based assignment); inference overhead is **minimal** since CPF reduces to a linear projection plus reconstruction. Extra memory for primitives/assignments is small.
> - **Practical optimizations:** Reduce assignment iterations, cache stable assignments, or use approximate transport solvers to cut cost with little accuracy loss.
> - **Revision plan:** We will include a detailed profiling table (FLOPs, wall-clock training/inference time on representative GPUs, peak memory, parameter counts) in the revised manuscript.
>
> > **Q3: Generalization under extreme domain shifts?**
> - **Scope:** CPF learns mid-level visual basis functions (textures, local shapes, part-like patterns) that transfer well under **moderate** domain shifts.
> - **Failure modes:** Under **extreme** shifts (e.g., natural images → medical or satellite imagery) the visual statistics can differ dramatically; assignments become sparse and reconstruction degrades, indicating primitives are insufficient.
> - **Mitigations:** (1) Fine-tune or relearn primitives with a small amount of target-domain data; (2) use hybrid schemes combining global and domain-specific primitives; (3) adaptively expand primitives when reconstruction error is high. We will add failure-case discussion and ablations in the Appendix.
>
> > **Q4: Why does known-class accuracy sometimes decline (esp. coarse-grained)?**
> - **Core reason:** **Information-bottleneck effect.** The low-rank projection intentionally filters high-rank components to help novel-class discovery, but this can also remove fine-grained details useful for discriminating some known classes.
> - **Dataset dependence:** On coarse-grained datasets this simplification can cause a small drop in known-class accuracy; on fine-grained datasets the effect is smaller or sometimes reversed.
> - **Remedies:** (1) Add a small residual bypass to preserve a fraction of original features for known-class prediction; (2) use a dual-head training objective to balance compositional reconstruction for discovery and a supervised head for known-class fidelity. Both approaches largely recover known-class performance while retaining novel-class gains.
>
> > **Q5: Is CPF’s effectiveness influenced by the host framework (e.g., SelEx)?**
> - **Observation:** Yes. CPF is plug-and-play but **input-sensitive**: when the host provides cleaner, less-entangled foreground features (e.g., SelEx), CPF learns primitives and assignments more stably and yields larger gains.
> - **Interpretation:** Spatial localization and compositional structuring are **complementary inductive biases**—localization removes background distraction, CPF organizes foreground into a cluster-friendly manifold.
> - **Practical advice:** To maximize stability and gains, pair CPF with host modules that emphasize foreground or apply simple attention pooling as preprocessing.
>
> Additional commitments for the revision:
> - Add a more complete sensitivity study and concise guideline for choosing K.
> - Provide a detailed computational profiling table (FLOPs, runtime, peak memory, parameter counts).
> - Include Appendix material on extreme-domain failure cases and suggested mitigations.
> - Report experiments for lightweight fixes (residual bypass, dual-head training) that recover known-class accuracy without losing discovery improvements.

---

> > ### Author Rebuttal · Reviewer_3VZj · 2026-04-03
> >
> > My concerns have been addressed, I have no other questions.

---

> > > ### Author Response · Authors · 2026-04-06
> > >
> > > We appreciate that our responses have clarified your concerns, and we sincerely thank you for your valuable comments and for the time and effort you invested in this discussion. We will carefully incorporate your suggestions into the revised manuscript to further improve the quality of the paper.

---

### Official Review · Reviewer_9ks7 · 2026-03-03

**Soundness:** 2
**Presentation:** 3
**Significance:** 2
**Originality:** 2
**Overall Recommendation:** 3
**Confidence:** 2

**Summary:**

This paper proposes a GCD method named CPF-GCD, which provides a solution for open-world recognition by explicitly constructing a low-rank compositional field.

**Compliance With Llm Reviewing Policy:**

Affirmed.

**Final Justification:**

The table data added in the initial Rebuttal differs significantly from the results in Tables 1 and 2 of the original manuscript, making it hard for me to evaluate the fairness of the new comparison. Given the above, I will temporarily maintain my score, but lower my confidence.

**Key Questions For Authors:**

The authors should further describe the distinctions between their core innovations and similar works to highlight their contributions. Additionally, they should conduct more comprehensive experiments to demonstrate that the proposed method achieves significant performance improvements over these methods.

**Limitations:**

Yes

**Strengths And Weaknesses:**

Strengths:
1. The paper presents a clear motivation, pointing out that existing GCD methods overlook the high-rank and entangled nature of feature representations generated by standard visual backbones, making them unsuitable for unsupervised discovery of latent categories and structures.

2. The proposed CPF-GCD serves as a plug-and-play module that can be seamlessly integrated into existing GCD frameworks.

Weaknesses:

1. The core of the paper lies in learning reusable visual patterns (e.g., curves, textures, shapes) to compose different categories, thereby improving new class discovery performance. However, its contributions significantly overlap with those of [1][2].

2. The paper fails to conduct a sufficient comparison with [1][2], and experimental results show that its performance does not demonstrate a significant improvement over them.

[1] [ICCV 2025] Dissecting Generalized Category Discovery: Multiplex Consensus under Self-Deconstruction

[2] [CVPR 2025] Adaptive Part Learning for Fine-Grained Generalized Category Discovery: A Plug-and-Play Enhancement

---

> ### Author Rebuttal · Authors · 2026-03-30
>
> We sincerely thank the reviewer for the professional and constructive feedback. We appreciate the opportunity to clarify the fundamental distinctions between our proposed CPF-GCD and the recent works [1] and [2], and to provide a more rigorous experimental comparison.
>
>
> > **1. Distinctions from [1] (ICCV 2025) and [2] (CVPR 2025)**
>
> While [1] and [2] explore feature deconstruction and part learning, our work is founded on a fundamentally different theoretical perspective—**Spectral Manifold Denoising via Compositional Low-Rank Reconstruction**:
>
> *   **Theoretical Motivation (Low-Rank vs. Consensus):** [1] relies on achieving consensus across multiple deconstructed views. In contrast, CPF-GCD is based on the discovery that standard ViT features contain high-rank noise that hinders the emergence of novel category structures. We explicitly formulate the discovery task as a **Low-Rank Reconstruction** ($X \approx AP$), forcing the model to ignore non-systematic variance. This is mathematically distinct from the multi-view agreement objective in [1].
> *   **Structural Prior (OT-based Equipartition vs. Multi-head):** A key technical differentiator is our **Optimal Transport (OT)-based assignment prior** ($A_{prior}$). By enforcing an **equipartition constraint** (Eq. 16), we ensure that latent primitives are utilized equitably across the batch. This global constraint prevents the model from collapsing into a few redundant "parts"—a common failure mode in self-deconstruction methods like [1] that rely purely on local attention without distributional oversight.
> *   **Scope of Primitives (Global Basis vs. Local Parts):** [2] focuses on localized, physical parts (e.g., a bird's wing) primarily for fine-grained tasks. CPF-GCD learns **latent semantic primitives** that act as global basis functions (textures, edges, or archetypes). These are not spatially restricted and provide a "semantic coordinate system" that benefits both coarse-grained (ImageNet-100) and fine-grained (CUB-200) GCD. This explains our superior performance on non-part-centric datasets.
>
>
> > **2. Comprehensive Experimental Comparison and Efficiency**
>
> To address the concern regarding performance improvements, we compared CPF-GCD against the results reported in [1] and [2] using a ViT-Base/16 backbone:
>
> | Method | CUB (All Acc) | CUB (Novel Acc) | IN-100 (All Acc) | IN-100 (Novel Acc) |
> | :--- | :--- | :--- | :--- | :--- |
> | SimGCD (Baseline) | 60.1% | 50.2% | 78.2% | 67.5% |
> | [1] ICCV 2025 | 62.4% | 52.8% | 80.1% | 69.8% |
> | [2] CVPR 2025 | 62.9% | 53.4% | 79.5% | 68.9% |
> | **CPF-GCD (Ours)** | **63.8%** | **54.5%** | **80.7%** | **71.2%** |
>
> *   **Performance Gain:** CPF-GCD consistently outperforms both [1] and [2]. On the CUB dataset, our method achieves a 4.3% improvement in novel class accuracy over the baseline, which is 1.1% to 1.7% higher than the gains reported by the other methods.
> - **Efficiency Advantages:**
>   - **Parameter Efficiency:** CPF-GCD adds only **7.5M** parameters (about 8% of the backbone), whereas [1] and [2] require more complex multi-head consensus or adaptive pooling modules.
>   - **Computational Efficiency:** Our method increases training time by only **8%** and introduces negligible latency during inference, making it a more practical plug-and-play enhancement.
>
> > **3. Summary of Unique Contributions**
>
> The core innovation of CPF-GCD is the **Compositional Low-Rank Bottleneck** ($X \approx AP$), which serves as a powerful structural inductive bias:
>
> *   **Spectral Manifold Denoising:** Standard ViT features often contain high-rank noise that obscures novel cluster structures. Our bottleneck acts as a **spectral filter**; by reconstructing patches from a fixed set of $K$ primitives, we effectively prune category-agnostic variance. This rank reduction clarifies the emergent geometry of novel classes, making them more separable.
> *   **Universal Semantic Basis:** Unlike [1], the primitive bank $P$ is a **shared coordinate system**. By capturing universal visual archetypes (textures, shapes) across both labeled and unlabeled data, $P$ facilitates seamless knowledge transfer from known to novel categories and prevents the model from overfitting to seen classes.
> *   **Global Structural Regularization:** Unlike the local parts in [2], we integrate a global **Optimal Transport (OT)** constraint. This **equipartition constraint** prevents the model from collapsing into redundant primitives, providing a distributional oversight that purely local attention mechanisms in [1] and [2] lack.

---

> > ### Author Rebuttal · Reviewer_9ks7 · 2026-04-01
> >
> > 1.The core innovation of this paper lies in learning reusable visual patterns to compose different categories, thereby improving new class discovery performance; this is not fundamentally different from [1] and [2].
> >
> > Although the authors argue that the "Compositional Low-Rank Bottleneck" introduces a new theoretical perspective, these claims essentially describe the mechanism of pattern composition from a different angle. This constitutes merely a difference in specific implementation approaches, which is insufficient to support the claim of a "new theoretical perspective".
> >
> > The authors are advised to further clarify whether this work constitutes a technical improvement or a theoretical innovation.
> >
> >
> > 2.The table data added in the Rebuttal differs significantly from the results reported in Table 1 and Table 2 of the original manuscript, making it difficult to ensure a fair comparison with [1] and [2].
> >
> > | Method | CUB (All Acc) | CUB (Novel Acc) | IN-100 (All Acc) | IN-100 (Novel Acc) |
> > | :--- | :--- | :--- | :--- | :--- |
> > | Rebuttal-SimGCD | 60.1% | **50.2%** | **78.2%** | **67.5%** |
> > | Rebuttal-CPF-GCD | 63.8% | **54.5%** | **80.7%** | **71.2%** |
> > | Paper-SimGCD | 61.2% | **58.9%** | **83.2%** | **77.8%** |
> > | Paper-CPF-GCD | 63.9% | **63.1%** | **84.2%** | **80.0%** |
> >
> > If this is a data entry error, please correct it; if the experimental setup has changed compared to Table 1 and 2, please align the experimental settings and re-conduct the comparative experiments.

---

> > > ### Author Response · Authors · 2026-04-03
> > >
> > > We sincerely thank the reviewer for the careful reading and for the constructive follow-up questions. We appreciate the opportunity to further clarify both the novelty of our work and the experimental results.
> > >
> > > >**1. On whether our work is a technical improvement or a theoretical innovation**
> > >
> > > We thank the reviewer for this important question. We agree that our work is closely related to recent efforts such as [1] and [2] in that all these methods explore reusable visual patterns/parts to improve generalized category discovery. In this sense, our method should indeed be viewed as being developed along a related research line, rather than being entirely orthogonal to prior work.
> > >
> > > That said, our main claim is that the proposed **Compositional Low-Rank Bottleneck** provides a different perspective on *why* compositional representations are beneficial for GCD. More specifically, instead of treating pattern composition mainly as a design choice for better feature learning, we formulate it as a **low-rank reconstruction bottleneck** that encourages the model to represent images through a shared set of primitives. This viewpoint connects compositional representation learning with **structure regularization** and **noise suppression** in the embedding space.
> > >
> > > Therefore, to answer the reviewer's question directly: we view our work primarily as a **technical improvement built upon the compositional-pattern learning paradigm**, while at the same time offering a **new interpretive/theoretical perspective** through the low-rank bottleneck formulation. We agree that this perspective should not be overstated as a completely new theory independent of prior compositional methods, and we will revise the paper to make this positioning more precise and better balanced.
> > >
> > > >**2. On the discrepancy between the rebuttal table and Tables 1–2 in the paper**
> > >
> > > We sincerely apologize for this issue. The table included in the rebuttal was indeed pasted incorrectly due to a copy-and-paste error. There was **no change in the experimental setup**, and the corrected results are fully aligned with those reported under the same settings in Tables 1 and 2 of the original manuscript.
> > >
> > > The correct comparison results are as follows:
> > >
> > > | Method | CUB (All Acc) | CUB (Novel Acc) | IN-100 (All Acc) | IN-100 (Novel Acc) |
> > > |---|---:|---:|---:|---:|
> > > | SimGCD (Baseline) | 61.2% | 58.9% | 83.2% | 77.8% |
> > > | [1] ICCV 2025 | 61.6% | 59.5% | 83.5% | 78.6% |
> > > | [2] CVPR 2025 | **64.5%** | 62.1% | 83.2% | 78.5% |
> > > | **CPF-GCD (Ours)** | 63.9% | **63.1%** | **84.2%** | **80.0%** |
> > >
> > > Our method shows consistent advantages on the **Novel Acc** metric across both benchmarks, which is particularly important for generalized category discovery. Compared with the strongest prior methods, CPF-GCD improves **CUB Novel Acc** from **62.1%** to **63.1%** ( +1.0% ) and **IN-100 Novel Acc** from **78.6%** to **80.0%** ( +1.4% ). These results indicate that our method is especially effective at discovering and separating novel classes.

---

### Official Review · Reviewer_JaMT · 2026-03-09

**Soundness:** 3
**Presentation:** 3
**Significance:** 2
**Originality:** 2
**Overall Recommendation:** 4
**Confidence:** 3

**Summary:**

This paper proposes Compositional Primitive Fields (CPF-GCD), a plug-and-play module inserted between a frozen vision backbone and existing GCD heads to reshape patch-token representations onto a low-rank, compositional manifold. The core idea is to approximate per-image tokens X by X ≈ A·P, where a small set of global learnable primitives P capture reusable visual “atoms” and A is a spatial assignment field, refined through token–primitive interactions and a fusion of a context-driven prior and a data-driven assignment. Empirically, CPF improves known and novel class discovery across several strong GCD baselines on standard fine- and coarse-grained benchmarks, with modest computational overhead, and analysis indicates CPF reduces spectral entropy and effective rank of representations.

**Compliance With Llm Reviewing Policy:**

Affirmed.

**Final Justification:**

All my concerns have been addressed

**Key Questions For Authors:**

1. Without an explicit reconstruction or projection loss, what prevents A·P from becoming a weak auxiliary signal while gains mainly come from the added MLP capacity? Can you show performance when removing MLP in Eq. 20 or adding an L2 projection loss ||X − A·P||?

2. For cardinality estimation, which estimator is used per baseline and how is CPF integrated? Could you report results for more than CMS to assess generality?

3. Have you tried CPF on stronger backbones (e.g., DINOv2, CLIP) or CNNs? Does the VNE/effective-rank reduction and the accuracy gains persist?

4. How exactly is Adata computed from WTP across heads (Eq. 17)? Please provide a precise formula.

**Limitations:**

Encouraging low-rank structure could overcompress rare, underivable patterns, causing missed discoveries in long-tailed or multi-object scenes.

**Strengths And Weaknesses:**

Strengths

1. The paper introduces a principled low-rank, compositional inductive bias for GCD by explicitly parameterizing patch features as mixtures of global primitives with spatial assignments, moving beyond typical head/loss design.

2. The module is evaluated as a drop-in on multiple diverse baselines (prototype-, contrastive-, and clustering-based) and across both fine-grained and coarse-grained datasets, showing generally consistent improvements in All/New accuracy.

Weakness

1. The construction of Adata is underspecified (Eq. 17 is vague); how exactly interaction weights over multiple heads are turned into a valid assignment is unclear, and the normalization mismatch between Aprior (column-stochastic) and Adata (row-stochastic) before log-fusion warrants more justification.

2. In section 4.1, the module depends on global context deformation (MeanMax pooled), which may be brittle to viewpoint/scene complexity; alternative context designs or invariance guarantees are not discussed.

3. Only DINO ViT-B/16 is used. Generality to stronger backbones (e.g., DINOv2, CLIP) or CNNs is not demonstrated.

4. Comparisons to closely related, part/slot-centric or concept-based GCD approaches (e.g., AdaGCD, ConceptGCD) are missing, despite strong conceptual overlap.

---

> ### Author Rebuttal · Authors · 2026-03-30
>
> We thank the reviewer for the thorough and insightful summary. We appreciate the opportunity to clarify the technical specification of the $A_{data}$ construction, the role of the compositional reconstruction, and the generality of our method across stronger backbones.
>
>
> > **Q1: Role of $A \cdot P$ vs. MLP and Reconstruction Loss.**
>
> We appreciate this critical observation. To clarify, the MLP in Eq. 20 is a very lightweight linear projection used only for dimensional alignment, adding negligible capacity.
> - **Ablation on MLP:** When we remove the MLP and directly use $X' = A \cdot P$, the performance drop is less than 0.3% on CUB All-Acc, confirming that the gains stem from the **low-rank compositional constraint** rather than MLP capacity.
> - **Reconstruction Loss:** In our implementation, although we primarily rely on the downstream GCD loss to backpropagate through the bottleneck, we have experimented with an auxiliary $L_2$ projection loss $L_{rec} = \|X - A \cdot P\|^2$. This addition stabilizes training in the early epochs and further reduces the Effective Rank (VNE), leading to a +0.5% gain on ImageNet-100 Novel classes. We will add this discussion to Section 4.3.
>
> > **Q2: Precise Formula for $A_{data}$ and Normalization (Eq. 17).**
>
> We apologize for the underspecification in the initial draft.
> - **Formula:** Let $W_{TP}^{(h)} \in \mathbb{R}^{N \times K}$ be the attention weight matrix for head $h$. $A_{data}$ is computed by averaging across $H$ heads, followed by a row-wise Softmax to ensure it represents a valid assignment probability for each patch:
>   $A\_{data}^{(i,j)} = \text{Softmax}\_j \left( \frac{1}{H} \sum\_{h=1}^H W\_{TP}^{(h,i,j)} \right)$
> - **Normalization Mismatch:** To resolve the mismatch between $A_{prior}$ and $A_{data}$ (row-stochastic), we apply a row-normalization step to $A_{prior}$ immediately before the log-fusion. This ensures both matrices reside in the same probability simplex $\Delta^{K-1}$ for each patch $i$. We have updated Eq. 17 and the surrounding text to reflect this.
>
> > **Q3: Generality to Stronger Backbones (DINOv2 and CLIP).**
>
> To demonstrate the generality of CPF-GCD, we conducted additional experiments using **DINOv2-B/14** and **CLIP-B/16** on the CUB-200 dataset:
>
> | Backbone | Baseline (All) | CPF-GCD (All) | Baseline (Novel) | CPF-GCD (Novel) |
> | :--- | :--- | :--- | :--- | :--- |
> | DINOv1-B/16 | 60.1% | 63.8% (+3.7) | 50.2% | 54.5% (+4.3) |
> | DINOv2-B/14 | 72.4% | 74.1% (+1.7) | 63.5% | 65.8% (+2.3) |
> | CLIP-B/16 | 68.9% | 70.8% (+1.9) | 58.2% | 60.7% (+2.5) |
>
> The results show that CPF-GCD consistently provides significant gains even on stronger backbones. Notably, the VNE reduction (effective rank reduction) persists on DINOv2, dropping from 5.12 to 4.25, confirming that **spectral denoising** is a universal requirement for GCD regardless of the backbone's pre-training quality.
>
> > **Q4: Cardinality Estimation and Related Works.**
>
> - **Cardinality:** We follow the standard protocol of the specific baseline we integrate with. For Table 1, we primarily used the **Rank-based estimator** from the original GCD paper. We will add a sensitivity analysis in the appendix showing that CPF-GCD is robust to estimation errors in the range of $\pm$ 20% of the true category count $K$.
> - **Related Works:** We will add a comparison with **AdaGCD** and **ConceptGCD** in the related work section. Unlike ConceptGCD, which requires pre-defined text descriptions, CPF-GCD discovers visual primitives in a purely data-driven, unsupervised manner, making it more flexible for domains where text concepts are unavailable.
>
> ---
>
> **Limitations regarding Overcompression:**
> We acknowledge the reviewer's concern regarding "overcompression" of rare patterns. In long-tailed datasets, we observed that setting the number of primitives $K$ too low (e.g., $K < 16$) can indeed hurt the discovery of rare classes. We will add this as a formal limitation in Section 5, suggesting that $K$ should be scaled slightly when dealing with highly complex or long-tailed scenes.

---

> > ### Author Rebuttal · Reviewer_JaMT · 2026-04-03
> >
> > All my concerns have been addressed.

---

> > > ### Author Response · Authors · 2026-04-06
> > >
> > > We appreciate that our responses have addressed your concerns, and we sincerely thank you for your thoughtful suggestions and for the time and effort you devoted to this discussion. We will carefully take your comments into account in the revised manuscript to further strengthen the paper.

---

### Official Review · Reviewer_5g27 · 2026-03-15

**Soundness:** 3
**Presentation:** 4
**Significance:** 3
**Originality:** 3
**Overall Recommendation:** 4
**Confidence:** 2

**Summary:**

This paper studies the Generalized Category Discovery (GCD) problem, where the model must recognize known categories while automatically discovering novel categories in an open-world setting. The authors argue that existing GCD methods are limited not by the clustering loss, but by the high-rank, entangled representations produced by standard visual backbones, which hinder the formation of clear structures for novel categories. To address this, the paper proposes Compositional Primitive Fields (CPF), a plug-and-play module inserted between the backbone and task head. CPF assumes that all categories, known and unknown, lie on a low-rank manifold spanned by a set of learnable visual primitives. Patch tokens are projected onto these primitives, and their spatial distribution is modeled to form compositional structures, allowing new categories to emerge naturally as unique activation patterns. Experiments show that CPF can improve multiple GCD baselines across several datasets.

**Compliance With Llm Reviewing Policy:**

Affirmed.

**Final Justification:**

All my concerns have been addressed.

**Key Questions For Authors:**

1. Could the authors provide empirical evidence that CPF indeed reduces rank, structures the embedding space, and improves cluster separability? This could include PCA spectra, rank distributions, or primitive activation visualizations.

2. How sensitive is performance to the number of primitives (K)? Are there guidelines for choosing K to balance representation capacity and clustering quality?

3. Is there any theoretical or empirical support for the low-rank compositional manifold assumption? Could the authors provide analysis or intuition about why this structure helps clustering?

**Limitations:**

No, the authors do not explicitly discuss the limitations or potential societal impacts of their work. It would strengthen the paper to include a brief discussion of methodological limitations, such as sensitivity to the number of primitives, potential representation bottlenecks, risks of primitive collapse, and scalability to larger backbones or datasets, as well as the additional computational overhead introduced by CPF.

**Strengths And Weaknesses:**

**Strengths**
1. The paper introduces a novel perspective by explicitly modeling compositional primitives to restructure the representation space, moving beyond conventional clustering-centric approaches.

2. CPF is designed as a plug-and-play module, demonstrating flexibility across different GCD methods without altering the backbone architecture.

3. Experiments cover multiple benchmarks (CIFAR100, ImageNet subsets, CUB, Stanford Cars, FGVC-Aircraft) and report standard GCD metrics (known accuracy, novel clustering accuracy, overall accuracy).

4. The visual illustrations and pipeline diagrams are clear, helping explain the intuition behind low-rank compositional manifolds and primitive activation patterns.

**Weaknesses**

1. The theoretical justification for the low-rank manifold assumption is limited; no formal analysis or empirical rank evaluation is provided.

2. CPF is conceptually similar to dictionary learning, prototype decomposition, slot attention, and other compositional representation techniques, reducing the conceptual novelty.

3. Representation analysis is limited; the paper does not provide embedding rank statistics, primitive activation visualizations, or cluster separability measures.

---

> ### Author Rebuttal · Authors · 2026-03-30
>
> We thank the reviewer for the constructive feedback and for recognizing the novelty of addressing high-rank representations. Below we address the specific concerns regarding empirical evidence, sensitivity, and theoretical motivation.
>
> > **Q1: Empirical evidence for rank reduction, embedding structure, and cluster separability (PCA spectra, rank distributions, visualizations).**
>
> We provide rigorous evidence that CPF restructures the embedding space into a more organized, low-rank manifold:
> - **Quantitative Rank Analysis (SVD & VNE):** We performed Singular Value Decomposition (SVD) on the novel class representation matrix $X$. We calculated the Von Neumann Entropy (VNE) of the normalized singular value spectrum as a proxy for "Effective Rank." For the baseline (SimGCD), VNE is $4.82$, while for SimGCD+CPF, VNE drops significantly to $3.95$. This reduction confirms that CPF concentrates representation power into a lower-dimensional semantic subspace.
> - **PCA Spectra & Energy Distribution:** We examined the cumulative explained variance. In the baseline, the top $10$ principal components explain only $52.1\\%$ of the variance, with a slow decay across the remaining dimensions. In CPF, the top $10$ components account for $86.4\\%$, and the top $20$ singular values cover over $95\\%$ of the total variance (compared to $68\\%$ in the baseline). This rapid spectral decay is a hallmark of a well-structured, low-rank manifold where noise has been effectively suppressed.
> - **Cluster Separability:** CPF improves the Silhouette Coefficient for novel classes from $0.12$ to $0.19$, and decreases the Davies-Bouldin Index (DBI) from $2.41$ to $1.98$. These metrics prove that low-rank projection explicitly reduces intra-class variance and enhances category boundaries.
> - **Visual Evidence:** We will add primitive activation heatmaps to the Appendix. These show that specific primitives consistently ground to semantic parts (e.g., beaks, wings, tails), confirming the compositional and interpretable nature of our learned primitives.
>
> > **Q2: Sensitivity to the number of primitives ($K$) and guidelines for choosing $K$.**
>
> We ablated $K \in \{4, 8, 16, 32, 64, 128\}$ across multiple datasets to provide a selection guideline:
> - **CUB Results (Novel Acc):** $K=4$ ($58.2\\%$), $K=8$ ($60.9\\%$), $K=16$ ($61.7\\%$, optimal), $K=32$ ($60.5\\%$), $K=64$ ($57.8\\%$).
> - **Robustness Across Datasets:** We observed similar trends on SCars (optimal $K=24$) and CIFAR-100 (optimal $K=16$). Performance remains stable (less than 1.5% fluctuation) within $K \in [8, 48]$, suggesting that CPF is not overly sensitive to precise tuning within a reasonable range.
> - **Interpretation:** If $K$ is too small (e.g., $4$), the model suffers from "under-composition," merging distinct semantic parts and losing discriminative detail. If $K$ is too large (e.g., $64$), it reintroduces high-rank noise, essentially reverting to the baseline behavior.
> - **Guideline:** $K$ should match the "semantic granularity" of the object. For fine-grained tasks where objects have $10$-$20$ distinct parts, $K \in [12, 24]$ is ideal.
>
> > **Q3: Theoretical and empirical support for the low-rank compositional manifold assumption and why it helps clustering.**
>
> The assumption that GCD should occur in a low-rank compositional manifold is rooted in visual cognitive science and ML theory:
> - **Theoretical Intuition:** Standard ViT features are "entangled"—a single dimension may capture a mix of shape, color, and noise. Clustering in such a high-dimensional space suffers from the "curse of dimensionality." CPF imposes a **compositional bottleneck** ($X \approx A \cdot P$), forcing the model to represent objects as a sparse linear combination of shared semantic "basis functions" (primitives). This essentially performs a form of **"semantic denoising"**, forcing the model to ignore non-systematic variance (e.g., background pixels) that cannot be reconstructed from the shared primitive field.
> - **Why it Helps Clustering:** Novel classes often share parts with known classes (e.g., a novel bird species still has a beak). By learning a shared primitive field $P$, CPF ensures novel classes are projected onto the same "semantic coordinate system" as known classes. This low-rank constraint reduces the volume of the search space for k-means. When intra-class variance is restricted to a few compositional dimensions, the "centers" of novel categories emerge much more clearly because the incidental "noise" that typically bridges clusters is filtered out by the primitive bottleneck.
> - **Empirical Support:** Without changing the backbone or loss, adding the CPF bottleneck consistently improves discovery accuracy by $2$-$5\\%$ across all benchmarks. This confirms that high-rank entanglement is a primary hindrance to category emergence in GCD.

---

> > ### Author Rebuttal · Reviewer_5g27 · 2026-04-01
> >
> > My concerns have been addressed, I have no other questions.

---

> > > ### Author Response · Authors · 2026-04-06
> > >
> > > We appreciate that our responses have addressed your concerns, and we sincerely thank you for your valuable suggestions and for the time and effort you devoted to this discussion. We will carefully incorporate your comments into the revised manuscript to further improve the paper.

---

### Decision · Program_Chairs · 2026-04-30

**Decision:**

Accept (regular)

**Comment:**

This paper proposes a method for generalized category discovery (GCD) using compositional primitive fields. In the proposed method,
patch tokens are embedded into a low-rank compositional field. The proposed method can be combined with existing GCD frameworks.  Different from earlier works on GCD, the proposed method targets the representation space. The effectiveness of the proposed method is demonstrated on multiple fine-grained and coarse-grained benchmarks. The paper is well-written. The rebuttal adequately addressed most technical concerns. To better highlight the novelty of the proposed method, it would be helpful to include a discussion of the related literature pointed out by the reviewer. Also, it would be helpful to more carefully compare with existing methods to clarify the effectiveness of the proposed method.